# Green Optimization of Sesame Seed Oil Extraction via Pulsed Electric Field and Ultrasound Bath: Yield, Antioxidant Activity, Oxidative Stability, and Functional Food Potential

**DOI:** 10.3390/foods14213653

**Published:** 2025-10-26

**Authors:** Vassilis Athanasiadis, Marianna Giannopoulou, Georgia Sarlami, Eleni Bozinou, Panagiotis Varagiannis, Stavros I. Lalas

**Affiliations:** Department of Food Science and Nutrition, University of Thessaly, Terma N. Temponera Street, 43100 Karditsa, Greece; vaathanasiadis@uth.gr (V.A.); mgiannop@uth.gr (M.G.); gsarlami@uth.gr (G.S.); empozinou@uth.gr (E.B.); pvaragiannis@uth.gr (P.V.)

**Keywords:** response surface methodology, definitive screening design, non-thermal processing, oxidative stability, antioxidant capacity, green food processing, functional lipid quality, plant-derived edible oils

## Abstract

Sesame seed oil is a bioactive-rich lipid source, notable for lignans, tocopherols, and unsaturated fatty acids that underpin its antioxidant and cardioprotective properties. This study optimized two innovative, non-thermal extraction techniques—pulsed electric field (PEF) and ultrasound bath-assisted extraction (UBAE)—to maximize yield and preserve oil quality for functional food applications. A blocked definitive screening design combined with response surface methodology modeled the effects of energy power (*X*_1_, 60–100%), liquid-to-solid ratio (*X*_2_, 10–20 mL/g), and extraction time (*X*_3_, 10–30 min) on fat content, DPPH antiradical activity, and oxidative stability indices (Conjugated Dienes, CDs/Conjugated Trienes, CTs). UBAE achieved the highest fat yield—59.0% at low energy (60%), high *X*_2_ (20 mL/g), and short *X*_3_ (10 min)—while PEF maximized DPPH to 36.0 μmol TEAC/kg oil at high energy (100%), moderate *X*_2_ (17 mL/g), and short *X*_3_ (10 min). CDs were minimized to 19.78 mmol/kg (UBAE, 60%, 10 mL/g, 10 min) and CTs to 3.34 mmol/kg (UBAE, 60%, 12 mL/g, 10 min). Partial least squares analysis identified *X*_2_ and *X*_3_ as the most influential variables (VIP > 0.8), with energy–time interplay (*X*_1_ × *X*_3_) being critical for antioxidant capacity. Compared to cold-pressing and Soxhlet extraction, PEF and cold-pressing retained higher antioxidant activity (~19 μmol TEAC/kg) and oxidative stability (TBARS ≤ 0.30 mmol MDAE/kg), while Soxhlet—though yielding 55.65% fat—showed the poorest quality profile (Totox value > 560). Both non-thermal techniques can deliver bioactive-rich sesame oil with lower oxidative degradation, supporting their application in functional foods aimed at improving dietary antioxidant intake and mitigating lipid oxidation burden. PEF at high energy/short time and UBAE at low energy/short time present complementary, scalable options for producing high-value edible oils aligned with human health priorities. As a limitation, we did not directly quantify lignans or tocopherols in this study, and future work will address their measurement and bioaccessibility.

## 1. Introduction

Sesame (*Sesamum indicum* L.) is one of the oldest oilseed crops, cultivated for thousands of years across Asia and Africa, with Myanmar, Tanzania, India, Sudan, and China among the leading producers [1]. The seeds are highly valued for their nutritional richness, containing approximately 22% protein, 62% fat, and significant amounts of minerals and vitamins [2]. Sesame oil, extracted from these seeds, has attracted global attention due to its unique composition and broad spectrum of health-promoting properties, making it an important dietary component and a promising functional food ingredient [3].

The bioactive composition of sesame oil is dominated by lignans, tocopherols, phytosterols, and polyunsaturated fatty acids (PUFAs), which collectively contribute to its oxidative stability and therapeutic potential [4,5]. Lignans such as sesamin and sesamolin exhibit antioxidant, anti-inflammatory, hypocholesterolemic, and neuroprotective effects, while sesamol demonstrates chemopreventive and detoxifying activities [6,7]. Tocopherols, particularly γ-tocopherol (521–990 mg/kg), with α- and δ-isomers also present, act as potent antioxidants that reduce LDL oxidation, inhibit platelet aggregation, and lower cardiovascular risk [8]. The fatty acid profile is characterized by high levels of oleic acid (35–45%) and linoleic acid (30–50%), providing essential ω-3 and ω-6 fatty acids that serve as precursors of eicosanoids involved in immune and inflammatory regulation [9]. These bioactives underpin sesame oil’s reported benefits in cardiovascular protection, metabolic regulation, neuroprotection, and cancer prevention [10,11,12].

The growing consumer demand for natural, bioactive-rich oils has expanded the use of sesame oil in nutraceuticals, dietary supplements, and functional food formulations [13]. Its favorable lipid profile and antioxidant content align with dietary strategies aimed at reducing oxidative stress, supporting vascular health, and promoting healthy aging [14]. Functional foods are defined as products that provide benefits beyond basic nutrition, and sesame oil fits squarely within this category due to its demonstrated bioactivity and stability [15].

Extraction methods play a critical role in determining both the yield and quality of sesame oil. Conventional approaches include cold-pressing and solvent-based Soxhlet extraction. Cold-pressing is widely used because it preserves bioactive compounds and avoids chemical residues, but it often results in lower oil yields [16]. Cold-pressing has been reported to be used in combination with acidic and enzymatic pre-treatments for the extraction of pumpkin, terebinth and flaxseed oils [17], while other treatments such as blanching of olive fruits prior to cold-pressing has also been reported [18]. Soxhlet extraction, by contrast, maximizes yield through exhaustive solvent circulation, but the prolonged exposure to heat and solvents accelerates lipid oxidation and degrades thermolabile compounds, leading to diminished nutritional quality [19]. These limitations have driven the development of non-thermal and environmentally friendly extraction technologies.

Among these, pulsed electric field (PEF) and ultrasound-assisted extraction (UAE) have emerged as promising alternatives. PEF applies short bursts of high-intensity electric fields that induce electroporation, enhancing cell permeability and mass transfer without significant heating [20]. For example, PEF pre-treatment has been shown to improve the extraction of polyphenols from onion tissues by enhancing cell permeability without compromising compound stability [21]. UAE, on the other hand, relies on acoustic cavitation to disrupt cell walls, improving solvent penetration and accelerating extraction. It has been reported to significantly increase the recovery of total phenolics and proteins from sugar beet leaves compared to traditional methods [22]. Both methods reduce processing time, solvent use, and energy demand, aligning with the principles of green extraction [23]. However, optimization of parameters—such as field strength and pulse duration in PEF or sonication time and amplitude in UAE—is crucial to avoid over-processing, which may lead to degradation or oxidation of target molecules [24,25].

Recent advances in sesame genetics have also provided insights into the regulation of oil quality traits. Xu et al. [4] fine-mapped a major pleiotropic QTL, i.e., a genomic region of large effect influencing both sesamin and sesamolin content, thereby highlighting the genetic basis of lignan accumulation in sesame. Similarly, phytosterol variation in sesame has been linked to specific loci and candidate genes, as demonstrated by Wang et al. [26], who identified key regulators of β-sitosterol, campesterol, stigmasterol, and Δ5-avenasterol content. These findings underscore the importance of integrating biochemical, technological, and genetic perspectives when evaluating sesame oil quality.

Despite extensive knowledge of sesame oil’s composition and health benefits, systematic optimization of non-thermal extraction methods such as PEF and UAE—particularly in direct comparison with conventional techniques—remains limited [27]. Few studies have benchmarked these approaches side-by-side in terms of yield, antioxidant activity, oxidative stability, and nutritional indices. Addressing this gap is essential for guiding the selection of sustainable, high-efficiency extraction strategies that preserve both yield and functional quality.

The aim of the present study is therefore to investigate sesame oil produced by PEF and ultrasound bath-assisted extraction (UBAE), evaluating oil content, fatty acid composition, quality parameters, oxidative stability, and antioxidant activity. Results were benchmarked against cold-pressing and Soxhlet extraction. By integrating response surface methodology (RSM), partial least squares (PLS), and chemometric tools, this work provides a comprehensive assessment of how extraction method and parameter optimization influence both yield and health-relevant quality attributes. The findings are expected to inform the development of sustainable extraction strategies for functional food applications and to highlight the trade-offs between efficiency and nutritional preservation in edible oil processing.

## 2. Materials and Methods

### 2.1. Sesame Seeds Material

Sesame seeds (Sudanese cultivars) were purchased from a local market in Karditsa, Greece. After opening the package, the seeds were stored in airtight plastic containers under refrigeration (4 °C) until use. For each experiment, the seeds were freshly ground to minimize oxidation and ensure sample homogeneity.

### 2.2. Chemicals and Reagents

Ammonium iron (II) sulfate, trichloroacetic acid, hydrochloric acid (37%), thiobarbituric acid, gallic acid monohydrate, Folin–Ciocalteu reagent, petroleum ether, and glacial acetic acid were purchased from Panreac (Barcelona, Spain). Cyclohexane, n-hexane, 2,2-diphenyl-1-picrylhydrazyl (DPPH), methanol, and *p*-anisidine were obtained from Sigma-Aldrich (Burlington, MA, USA). Isooctane, ethyl acetate, and dichloromethane were supplied by Carlo Erba (Vaul de Reuil, France). Chloroform, anhydrous sodium carbonate, and ammonium thiocyanate were purchased from Penta (Prague, Czech Republic). Malondialdehyde and tocopherol standards were obtained from Merck Ltd. (Darmstadt, Germany). Trolox (6-hydroxy-2,5,7,8-tetramethylchroman-2-carboxylic acid) came from Glentham Life Sciences (Corsham, UK), and ethanol (99.8%) was obtained from Fischer Scientific (Loughborough, UK).

### 2.3. Experimental Design

To optimize the extraction of bioactive compounds from sesame seeds, this study used Response Surface Methodology (RSM) with a blocked Definitive Screening Design (DSD), applying both Pulsed Electric Field (PEF) and Ultrasound Bath-Assisted Extraction (UBAE) techniques as the blocking factor. The independent variables investigated included energy power (*E*, %), liquid-to-solid ratio (*R*, mL/g), and extraction time (*t*, min), each tested at three coded levels (−1, 0, +1). Preliminary experiments informed the selection of optimal ranges: 0.6–1.0 kV/cm for PEF electric field intensity and 132–220 W for UBAE ultrasonic power. In this design (Table 1), *E* (%) denotes instrument set points mapped to these physical ranges within each block (PEF and UBAE), so that the coded levels (−1/0/+1) correspond to the low/center/high actual settings per technique. Additional constants were used to standardize conditions—such as petroleum ether as the solvent (40 mL per run), pulse duration of 10 μs, and pulse period of 1000 μs for PEF; and a 37 kHz frequency in pulsed mode for UBAE. The solvent volume was fixed at 40 mL, and the sample mass was adjusted (2–4 g) to achieve the desired liquid-to-solid ratios. Across 18 experimental runs, balanced as 9 per block, each condition was triplicated to ensure reproducibility and statistical robustness. Vessel geometries were kept constant across all experiments to avoid mass-transfer bias.

These settings were derived from exploratory trials that revealed specific thresholds where compound recovery and DPPH antiradical activity were most pronounced. Lower and higher energy levels exhibited differential impacts on fat content and conjugated diene/triene profiles. Extraction time and solvent ratios also demonstrated notable influence on yield efficiency and oxidative stability. The preliminary observations helped refine each variable’s range to capture potential nonlinear behavior. With controlled energy inputs and consistent mechanical parameters, the study established a structured and reliable foundation for extracting and evaluating sesame seed bioactives through cutting-edge, synergistic extraction methods.

To ensure accurate prediction of experimental outcomes, we used stepwise regression with Bayesian Information Criterion (BIC) minimization to select a parsimonious second-order response surface model. Starting with the full quadratic form with all two-way interactions (and block-by-term effects for technique), variables were coded to −1, 0, +1 and coefficients were estimated by ordinary least squares. The resulting model captures the interplay among the three key independent variables and enables precise interpretation of their combined effects on each response. After data collection, a second-order polynomial response surface was fitted, with term selection performed via stepwise regression minimizing the BIC while preserving model hierarchy.(1)$$Y_{k}=\beta_{0}+\sum_{i\;=\;1}^{3}\beta_{i}X_{i}+\sum_{i\;=\;1}^{3}\beta_{\mathrm{ii}}X_{i}^{2}+\sum_{i\;=\;1}^{2}\sum_{j\;=\;i\;+\;1}^{3}\beta_{\mathrm{ij}}X_{i}X_{j}\;+\;\gamma \cdot \mathrm{Block}\;+\sum_{m}\delta_{m}\left(\mathrm{Block}\times {\mathrm{term}}_{m}\right)$$
where the independent variables are denoted by *X_i_* and *X_j_*, and the predicted response variable is defined by *Y_k_*. In the model, the intercept *β*_0_ and regression coefficients *β_i_*, *β_ii_*, and *β_ij_* represent the linear, quadratic, and interaction terms, respectively, while *γ* and *δ_m_* account for main and interaction effects of the extraction-technique block.

### 2.4. Determinations

#### 2.4.1. Oil Content

The oil content of the sesame seeds was determined gravimetrically after the evaporation of the solvent using a Heidolph Laborota 4000/G3 rotary evaporator, equipped with Rotavac Valve Control (Heidolph Instruments GmbH & Co. KG, Schwabach, Germany) for the samples treated with PEF, UBAE and Soxhlet and after centrifugation for the cold-pressed treated sample. The following formula was used:(2)$$\mathrm{Oil}\mathrm{Yield}(\%)=\frac{m_{1}-m_{2}}{m_{s}}\times 100$$
where $m_{1}$ is the mass of the empty extraction flask (g), $m_{2}$ is mass of the flask after solvent evaporation and drying (g) and $m_{s}$ is mass of the sample (grinded sesame seeds) used for extraction (g).

#### 2.4.2. DPPH^•^ Antiradical Activity

The antioxidant capacity against the 2,2-diphenyl-1-picrylhydrazyl radical (DPPH^•^) was determined following the procedure of Athanasiadis et al. [28], with minor modifications. In contrast to the original protocol, which combined 1 mL of a 10% (*w*/*v*) diluted oil sample with 4 mL of freshly prepared 100 μM DPPH^•^ solution in ethyl acetate, the present study employed 950 μL of 100 μM DPPH^•^ solution (ethyl acetate) mixed with 50 μL of a ten-fold diluted oil sample. Absorbance was recorded at 515 nm both immediately (*A*_515(i)_) and after 30 min (*A*_515(f)_) against a blank. The percentage inhibition of DPPH^•^ was calculated as follows:(3)$$\mathrm{Inhibition}\left(\%\right)=\;\left(\frac{A_{515(i)}-A_{515(f)}}{A_{515(i)}}\right)\times 100$$

Results were expressed as μmol Trolox equivalent antioxidant capacity (TEAC) per kilogram of oil to normalize antioxidant capacity per oil mass, consistent with edible oil reporting standards. The calibration curve prepared with Trolox (50–500 μM; y = 0.1566x + 0.6698, R^2^ = 0.999), and had LOD = 15.1 μM and LOQ = 45.74 μM. Measurements were performed in 1 cm quartz cuvettes.

#### 2.4.3. Conjugated Dienes and Trienes

The determination of conjugated dienes (CD) and trienes (CT) was carried out following the procedure of Pegg et al. [29]. Briefly, 0.01 g of oil was dissolved in cyclohexane in a 5 mL volumetric flask. Absorbance was measured at 232 nm and 270 nm for CD and CT, respectively. The concentrations were calculated using the following equations:(4)$$C_{\mathrm{CD}}(\mathrm{mmol}/\mathrm{mL})=\frac{A_{232}}{\varepsilon \times l}\;{\mathrm{CD}}_{\mathrm{value}}(\mathrm{mmol}/\mathrm{kg}\mathrm{Oil})=\frac{C_{\mathrm{CD}}\times (5\times 10^{3})}{w}$$
(5)$$C_{\mathrm{CT}}(\mathrm{mmol}/\mathrm{mL})=\frac{A_{270}}{\varepsilon \times l}\;{\mathrm{CT}}_{\mathrm{value}}(\mathrm{mmol}/\mathrm{kg}\mathrm{Oil})=\frac{C_{\mathrm{CT}}\times (5\times 10^{3})}{w}$$
where *ε* is the molar absorptivity of linoleic acid hydroperoxide (2.525 × 10^4^ M^−1^ cm^−1^) [29], *l* is the cuvette path length (1 cm), and 5 × 10^3^ accounts for the solvent volume (5 mL). CD and CT contents were expressed as mmol per kg of oil.

### 2.5. Benchmarking Method Specifications

For cold-pressed sesame oil, approximately 10 g of sesame seeds were ground and wrapped in a cotton fabric bag before being placed in a hydraulic press (Atlas Manual Hydraulic Press, Specac Ltd., Kent, UK). A force of roughly 10 tons was applied to squeeze out the oil, and the resulting sesame pulp was collected. The pulp was then transferred to centrifuge tubes and spun at 10,000 rpm, causing the remaining oil to separate into a clear upper phase. The oily layer was carefully decanted, weighed, and reserved for all subsequent analyses.

For Soxhlet extraction, another 10 g of ground sesame seeds were loaded into a thimble and extracted with the same solvent (petroleum ether) at a liquid-to-solid ratio (20 mL/g) matching the optimal conditions. The system ran continuously for 300 min (5 h), which is ten times longer than the optimal PLS-predicted duration. After extraction, the solvent was removed under reduced pressure using a rotary evaporator, and the residual oil was weighed and used in further assays.

#### 2.5.1. Peroxide Value (PV)

Peroxide values (PV) of sesame oil samples were determined according to the IDF standard method 74A:1991 with modifications described by Athanasiadis et al. [30]. Briefly, 0.05 g of oil was dissolved in 2 mL of dichloromethane/ethanol (3:2 *v*/*v*) in a 2 mL Eppendorf tube using a vortex mixer for 2–4 s. Then, 20 μL of the oil extract was mixed with 1960 μL of the same solvent. Next, 10 μL of 4 M ammonium thiocyanate solution was added and vortexed for 2–4 s, followed by 10 μL of 25.5 mM ammonium iron(II) sulfate in 10 M HCl and vortexing for another 2–4 s. After incubation for 5 min at room temperature, absorbance was recorded at 500 nm against a reagent blank (reaction mixture without oil) using a UV spectrophotometer. PV were determined from a hydrogen peroxide (H_2_O_2_) calibration curve constructed with six concentrations (50–500 μmol/L in dichloromethane/ethanol) and expressed as mmol H_2_O_2_ per kg of oil, according to(6)$$\mathrm{PV}\left(\mathrm{mmol}H_{2}O_{2}/\mathrm{kg}\mathrm{Oil}\right)=\frac{C_{H_{2}O_{2}}\times V}{w}$$
where *C*_H_2_O_2__ is the H_2_O_2_ concentration (in μmol/L), *V* is the volume of the extraction medium (L), and *w* is the mass of the oil sample (g).

#### 2.5.2. Thiobarbituric Acid Reactive Substances (TBARS)

The thiobarbituric acid reactive substances (TBARS) assay was performed following the method of Qiu et al. [31]. Briefly, 0.1 g of oil was mixed with 5 mL of TBA solution, shaken thoroughly, and incubated at 95 °C for 20 min. The TBA solution was prepared by dissolving 15 g of trichloroacetic acid, 1.76 mL of concentrated HCl, and 0.375 g of TBA in 100 mL of deionized water. Following incubation, the samples were cooled in an ice bath for 5 min, after which 200 μL of chloroform was added. The mixture was vortexed and centrifuged at 4500 rpm for 10 min. The absorbance of the resulting supernatant was measured at 532 nm using deionized water as a blank. TBARS values were determined from a malondialdehyde calibration curve (15–300 μmol/L; y = 0.0032x − 0.0004, R^2^ = 0.9999) and expressed as mmol malondialdehyde equivalents (MDAE) per kg of oil, according to(7)$${\mathrm{TBA}}_{\mathrm{value}}(\mathrm{mmol}\mathrm{MDAE}/\mathrm{kg}\mathrm{Oil})=\frac{C_{\mathrm{MDA}}\times V}{w}$$
where *C*_MDA_ is the malondialdehyde concentration (mmol/L), *V* is the volume of extraction medium (L), and *w* is the weight of the oil sample (g).

#### 2.5.3. *p*-Anisidine Value (*p*-AV)

The *p*-anisidine value (*p*-AV) of the oil samples was determined according to EN ISO 6885:2012 [32]. Briefly, 0.5 g of oil was dissolved in isooctane in a 10 mL volumetric flask. For the unreacted sample, 1 mL of this solution was mixed with 0.2 mL of glacial acetic acid, shaken vigorously, kept in the dark for 10 min, and the absorbance (*A*_0_) was measured at 350 nm. For the reacted sample, 1 mL of the diluted oil solution was combined with 0.2 mL of *p*-anisidine reagent (0.5% in glacial acetic acid), shaken, incubated for 10 min in the dark, and absorbance (*A*_1_) recorded at 350 nm. The reagent blank was prepared by mixing 1 mL of isooctane with 0.2 mL of *p*-anisidine reagent, shaken, incubated for 10 min in the dark, and absorbance (*A*_2_) measured at 350 nm. The *p*-AV was then calculated according to(8)$$p\text{-}\mathrm{AV}=\frac{100\;QV}{m}0.24[(A_{1}-A_{2}-A_{0})]=12(\frac{A_{1}-A_{2}-A_{0}}{m})$$
where *Q* is the sample concentration in solution (0.05 g/mL), *V* is the solution volume (10 mL), *m* is the mass of the test portion (g), and the correction factor of 0.24 accounts for the 20% dilution effect introduced by adding 0.2 mL of reagent or glacial acetic acid to 1 mL of oil solution, as specified in EN ISO 6885:2012 [32].

#### 2.5.4. Totox Value

The Totox value (TV), representing the overall oxidation of oil by combining primary and secondary oxidation products, was calculated following Galanakis et al. [33]. It was determined as the sum of the peroxide value (PV) and twice the *p*-anisidine value (*p*-AV):
(9)TV=2×PV+p-AV

#### 2.5.5. Fatty Acids Profile

Fatty acid methyl esters (FAMEs) were prepared from the oil samples according to Annex XB of Commission Regulation (EC) No. 796/2002 [34]. Gas chromatographic analysis was performed on an Agilent 7890A system (Agilent Technologies, Santa Clara, CA, USA) equipped with an Omegawax capillary column (30 m × 320 μm × 0.25 μm; Supelco, Bellefonte, PA, USA). Helium was used as the carrier gas at a flow rate of 1.4 mL/min. The oven temperature program was set as follows: initial isotherm at 70 °C for 5 min, ramped to 160 °C at 20 °C/min, increased to 200 °C at 4 °C/min, and finally raised to 240 °C at 5 °C/min. The injector and flame ionization detector (FID) were maintained at 240 °C and 250 °C, respectively. Hydrogen and air flow rates were 50 and 450 mL/min, respectively, with a helium makeup flow of 50 mL/min. Samples (1.0 μL) were injected in split mode (1:100). Fatty acid peaks were identified by comparison with a FAME Mix C8–C24 reference standard (Sigma-Aldrich, St. Louis, MO, USA). The relative percentage composition of fatty acids was calculated by the normalization method (without correction factors) based on GC peak areas, and values were averaged from triplicate GC-FID determinations.

#### 2.5.6. ATR–FTIR Spectroscopy

Fourier-transform infrared (FTIR) spectra were obtained using a Shimadzu Prestige 21 spectrophotometer (Shimadzu Corporation, Kyoto, Japan) equipped with an attenuated total reflectance (ATR) accessory fitted with a ZnSe trough plate. The system was configured with a DLATGS detector (deuterated L-alanine-doped triglycine sulfate), a KBr beam splitter, high-throughput optical components, and a ceramic light source. For each oil sample, 0.8 mL was applied directly to the ATR crystal, and 32 scans were collected over the spectral range of 4000–400 cm^−1^ at a resolution of 4 cm^−1^. The ATR crystal was cleaned with acetone before and after each run to prevent cross-contamination. All measurements were performed in triplicate, and spectra were processed using IRsolution software (version 1.60, Shimadzu, Kyoto, Japan) for baseline correction and signal-to-noise evaluation.

#### 2.5.7. Specific Energy Consumption for the PEF and UBAE Treatments

The specific energy consumption was calculated as follows [35,36]:(10)$$E_{\mathrm{specific}}\left(\mathrm{kJ}/\mathrm{kg}\mathrm{oil}\right)=\frac{P\times t}{m_{\mathrm{oil}}}$$
where *P* is the applied power (W or V × A), *t* is the treatment time (min) and *m*_oil_ is the mass of the extracted oil (kg).

### 2.6. Statistical Analysis

Each experimental condition was performed in triplicate (*n* = 3) within a blocked design, with the full set of extraction runs executed in duplicate as two independent batches. Response variables were reported as mean ± standard deviation. The distribution of residuals for each model was checked for normality using the Shapiro–Wilk test in JMP Pro 16 (SAS Institute Inc., Cary, NC, USA). One-way analysis of variance (ANOVA) was used, where appropriate, to assess statistically significant differences between extraction techniques or operating conditions, with significance accepted at *p* < 0.05. For multi-factor optimization, second-order polynomial response surface models were built using a stepwise selection procedure minimizing the Bayesian Information Criterion (BIC) and fitted by ordinary least squares (OLS) on coded variables; lack-of-fit was tested against pure error from replicates. Model adequacy was evaluated via R^2^, adjusted R^2^, RMSE, and other metrics. Multivariate chemometric tools—including Partial Least Squares (PLS) regression, Principal Component Analysis (PCA), and Pareto plot analysis—were applied to explore inter-relationships among process variables and quality indices, identify the most influential factors (VIP > 0.8), and visualize the optimization space. Two independent batches were executed; between-batch CVs were approximately 16% for fat yield, 24% for DPPH, 14% for CDs, and 17% for CTs. These systematic baseline differences were captured by the Block term in the models. Batch was therefore not included as a random effect, but results were consistent across replicates within each batch.

## 3. Results and Discussion

### 3.1. RSM Overview Across Design Space and Yield–Antioxidant Trade-Offs

Across the design space (Table 2), RSM revealed clear technique-dependent trends: UBAE generally achieved the highest fat yields, peaking at 59.0% under low energy, high liquid-to-solid ratio and short extraction time, while PEF excelled in antioxidant capacity, with DPPH values above 33 μmol TEAC/kg oil at high energy and short time; fat recovery in both methods was strongly enhanced by increasing solvent ratio, and in PEF also by longer extraction time, whereas energy effects were mixed, boosting DPPH under PEF only when exposure time was short. Oxidation markers showed complementary patterns: CDs and CTs were minimized by PEF within specific *X*_2_–*X*_3_ combinations (e.g., 18.3 mmol/kg oil CDs at (0, +1, +1)) and in UBAE at low-mid settings, but UBAE mid-range conditions tended to elevate CTs. These surfaces highlight trade-offs—UBAE’s yield gains often coincide with modest DPPH, and PEF’s antioxidant peaks occur without maximum fat recovery—allowing operators to target high yield, high antioxidant activity, or oxidative stability by tuning energy, solvent ratio, and time within each technique’s optimal window. Having outlined the overall design-space responses, we next examined the fitted models in detail to identify which coefficients and interactions were most influential. This transition emphasizes the thematic link between global trends and statistical interpretation.

### 3.2. Model Analysis and Key Coefficients

The fitted quadratic response surface models adequately described the effects of energy power (*X*_1_), liquid-to-solid ratio (*X*_2_), and extraction time (*X*_3_) on all measured responses (fat yield, DPPH activity, conjugated dienes, and conjugated trienes). All models were statistically significant (*p* < 0.01), with coefficients of determination (R^2^) ranging from 0.85 to 0.98 and adjusted R^2^ values from 0.74 to 0.94 (Appendix A), indicating strong predictive performance.

Among the main effects, *X*_2_ (liquid-to-solid ratio) consistently emerged as the dominant positive factor for fat yield and DPPH activity, whereas *X*_1_ (energy power) and *X*_3_ (extraction time) exerted more complex, response-specific influences. For antioxidant activity (DPPH), the *X*_1_ × *X*_3_ interaction was particularly strong, reflecting that high energy benefits are limited to shorter extraction times. For oxidation markers, negative *X*_1_ × *X*_3_ and *X*_2_ × *X*_3_ interactions indicated that combined increases in energy or solvent ratio with time can suppress further oxidation.

Significant blocking terms confirmed systematic differences between extraction techniques: UBAE generally showed higher baseline yield, while PEF achieved lower oxidation indices and preserved higher antioxidant capacity. Although lack-of-fit tests were significant for some responses (fat yield, DPPH, and CTs), the models reliably captured the main factor trends and interactions within the experimental domain.

Detailed polynomial equations, regression coefficients, and full ANOVA statistics—including lack-of-fit tests and model selection criteria—are provided in Appendix A.

The contour plots in Figure 1 clearly illustrate the technique-dependent response surfaces and how factor interactions shape each outcome. In Figure 1A, fat content rises markedly with increasing liquid-to-solid ratio (*X*_2_), with PEF (Block 1) also showing gains from longer extraction time (*X*_3_), whereas UBAE (Block 2) achieves broader high-yield regions less sensitive to time. Figure 1B shows that for DPPH activity, PEF surfaces peak at high energy power (*X*_1_) and short extraction time, with performance declining rapidly as *X*_3_ increases, while UBAE responses are generally lower and more affected by high *X*_2_ or prolonged *X*_3_; all three factor-pair combinations reveal meaningful curvature and interaction effects. In Figure 1C, CDs are minimized in PEF at higher *X*_1_ and shorter *X*_3_, and in UBAE by moving away from mid-range settings, with negative *X*_1_ × *X*_3_ and *X*_2_ × *X*_3_ interactions evident. Figure 1D indicates that CTs vary only modestly, tending to be lower at high *X*_2_ and low *X*_3_, with PEF surfaces consistently below UBAE across all factor-pair plots. Collectively, these figures confirm that PEF tends to produce steeper, more peaked surfaces—signifying greater sensitivity to operating conditions—while UBAE landscapes are flatter and more forgiving, but with more pronounced trade-offs between yield, antioxidant capacity, and oxidative stability. These trends are further illustrated in the supplementary 3D response surface plots (Appendix A), which provide an interactive visualization of how the combined effects of energy power, solvent ratio, and extraction time shape each response variable.

Figure 2 highlights where factors truly interact and how technique reshapes those effects: for fat content (plot A), lines are essentially parallel in *X*_1_ comparisons but diverge across *X*_2_–*X*_3_, confirming a pronounced *X*_2_ × *X*_3_ interaction—time boosts fat more when the liquid-to-solid ratio is high—with PEF showing steeper time sensitivity and UBAE broader, flatter gains. For DPPH (plot B), the strongest interaction is *X*_1_ × *X*_3_: energy raises DPPH only at short times, with benefits eroding as time increases; smaller positive *X*_1_ × *X*_2_ and *X*_2_ × *X*_3_ effects add curvature, and PEF displays greater slope changes than UBAE. For CDs (plot C), time tends to increase oxidation, but negative *X*_1_ × *X*_3_ and *X*_2_ × *X*_3_ interactions flatten that rise at higher energy or higher solvent ratio; PEF offsets the baseline downward relative to UBAE and moderates the time penalty more effectively. For CTs (plot D), interactions are subtle—most evident for *X*_2_ × *X*_3_ (negative)—so CTs are lowest when solvent ratio is high and time is short; technique mainly acts as a vertical shift, with PEF consistently lower. Together, the interaction plots show that PEF produces sharper, technique-dependent slope changes (especially along energy–time for DPPH and time–solvent for fat and CDs), while UBAE is more stable but tends to trade higher yields for lower antioxidant capacity and higher oxidation at mid-range settings. These two-dimensional interaction patterns are consistent with the three-dimensional RSM surfaces provided in the Appendix A, which visually confirm the same curvature and technique-dependent slopes observed in Figure 1 and Figure 2.

Table 3 reflects optima that are consistent with the patterns observed in your regression models, contour plots, and interaction analyses. The conditions yielding maximum predicted responses varied by target, underscoring the trade-offs between yield, antioxidant capacity, and oxidative stability. For fat content, the model points to UBAE at low energy (60%), high liquid-to-solid ratio (20 mL/g), and long extraction time (30 min) as optimal, producing ~48% fat with moderate desirability (0.698). In contrast, the DPPH optimum occurs under PEF at high energy (100%), intermediate solvent ratio (17 mL/g), and short time (10 min), achieving a high antiradical value (~36 μmol TEAC/kg oil) with near-perfect desirability (0.992), mirroring the strong *X*_1_ × *X*_3_ effect seen in plots. For minimizing CDs, UBAE at low energy, low solvent ratio, and short time (60%, 10 mL/g, 10 min) yields the lowest predicted oxidation (19.78 mmol/kg) with desirability 0.834. Similarly, the lowest CTs were predicted for UBAE at 60% energy, 12 mL/g, and 10 min (3.34 mmol/kg, desirability 0.889). The split between UBAE for yield/oxidative indices and PEF for antioxidant activity aligns with earlier findings: UBAE delivers higher fat recovery and can suppress oxidation under specific low-energy, short-time conditions, while PEF excels at preserving antioxidant potential under high-energy, rapid extraction. These optima make clear that a single set of parameters cannot simultaneously maximize all quality attributes, highlighting the need for multi-response optimization when both yield and quality are critical.

### 3.3. Factor Importance via Pareto Plots

To further clarify the relative importance of each factor, Pareto plots were employed. These highlight yield–antioxidant trade-offs and oxidative stability drivers in a single comparative framework. In Figure 3, the Pareto plots make it clear which factors and interactions most strongly shape each response. For fat content, the largest negative effect is the main block term for PEF, while the liquid-to-solid ratio (*X*_2_) and its interaction with block drive positive gains, confirming *X*_2_’s central role in yield. DPPH is dominated by energy power (*X*_1_) effects, with a strong positive quadratic term and block shift for PEF boosting activity, countered by significant negative impacts from extraction time (*X*_3_) and its interaction with *X*_1_. In CDs, significant negative *X*_1_ × *X*_3_ and *X*_2_ × *X*_3_ terms show that higher energy or solvent ratio can temper oxidation increases at longer times, alongside a general lowering of CDs in PEF. CTs display smaller but meaningful effects, with the block lowering values for PEF, a negative *X*_2_ × *X*_3_ interaction, and curvature in *X*_2_ and *X*_3_ indicating minima at high solvent ratio and short time. Collectively, the plots highlight *X*_2_’s consistent importance for yield, *X*_1_ and time-related interactions as key to antioxidant performance, and targeted factor combinations for managing oxidation indices.

### 3.4. Oxidative Stability Patterns from Principal Component Analysis (PCA)

Beyond univariate models, multivariate PCA was applied to capture oxidative stability patterns across all indices simultaneously, grouping samples according to technique and extraction conditions. The PCA biplot in Figure 4 visualizes how the three process variables (blue vectors: *X*_1_ = energy power, *X*_2_ = liquid-to-solid ratio, *X*_3_ = extraction time) relate to the four measured responses (red vectors) within each technique, and how these relationships structure the overall variability. For PEF (Block 1), PC1 (≈53% variance) and PC2 (≈27%) together capture about 80% of the dataset’s variance. Fat content, DPPH, CDs, and CTs load in different quadrants, with Fat aligning positively with both *X*_2_ and *X*_3_, consistent with earlier regression results showing those as key yield drivers. DPPH is closely associated with high *X*_1_ and short *X*_3_, while CDs and CTs cluster more with longer times, indicating oxidative indices rise along the time axis. For UBAE (Block 2), PC1 (≈49%) and PC2 (≈28%) explain a similar proportion of variance. Fat content aligns strongly with *X*_2_ and to a lesser extent *X*_3_, but the angle to DPPH is wider than in PEF, confirming the weaker antioxidant-yield coupling in this technique. CDs and CTs are positioned closer together and show greater correlation with *X*_3_, suggesting that time plays a common role in oxidation for UBAE. *X*_1_’s shorter vector length hints at a smaller overall contribution of energy setting to response separation compared with PEF. Overall, the biplots reinforce earlier findings: *X*_2_ is a dominant driver of yield in both methods, *X*_1_ has a stronger influence on antioxidant activity in PEF than in UBAE, and oxidative markers group with time-driven trends. The relative vector orientations highlight inherent trade-offs and technique-specific sensitivities, making PCA a useful complement to the contour and Pareto analyses for visualizing multi-response optimization space.

Table 4 shows clear technique-specific correlation structures. Under PEF (Block 1), fat is essentially decoupled from the other responses (r ≤ 0.24), while DPPH correlates positively and strongly with both CDs (r = 0.609) and CTs (r = 0.742), implying that the conditions elevating antiradical capacity also co-vary with higher oxidation markers in this technique. CDs and CTs are only weakly related (r = 0.233), suggesting different time/energy sensitivities for primary vs. secondary oxidation under PEF. Under UBAE (Block 2), trade-offs are more pronounced: fat correlates moderately negatively with DPPH (r = −0.478) and CDs (r = −0.470), and only weakly with CTs (r = 0.179), indicating that high-yield settings tend to depress antioxidant capacity yet are associated with lower primary oxidation. DPPH relates positively to CDs (r = 0.456) but slightly negatively to CTs (r = −0.192), pointing to a divergence between early-stage oxidation and propagation products in UBAE. Overall, PEF exhibits yield–quality decoupling with DPPH tracking oxidation indices, whereas UBAE shows clearer yield–antioxidant and yield-oxidation trade-offs—guiding distinct multi-response optimization strategies for each technique.

The differential tracking of conjugated dienes/trienes (CDs/CTs) by DPPH in PEF versus UBAE likely arises from the distinct energy-time profiles and cavitation effects of these methods. PEF delivers brief, high-intensity electric pulses that cause electroporation and rapid release of bioactives, often preserving their native antioxidant properties, while UBAE involves prolonged sonication where cavitation generates microbubbles and localized hotspots, potentially altering antioxidant molecule structures and thus impacting DPPH reactivity differently compared to actual lipid oxidation markers [37,38]. It is important to caution against using DPPH alone as a proxy for oxidative stability because DPPH, as an electron-transfer-based assay, may not reliably reflect changes in primary or secondary oxidation products under conditions where extraction-induced physical or chemical modifications impact radical scavenging capacity independently of true oxidative deterioration [39,40].

### 3.5. Integrated Optimization Through Partial Least Squares (PLS)

Partial Least Squares (PLS) regression was then used to integrate yield, antioxidant activity, and oxidation indices, providing a holistic view of trade-offs and enabling prediction of optimal conditions. The PLS prediction profiler in Figure 5A shows how overall desirability emerges from balancing the three levers: raising the liquid-to-solid ratio (*X*_2_) consistently lifts desirability via higher fat and lower CTs, energy power (*X*_1_) chiefly tunes DPPH with diminishing returns captured by curvature, and extraction time (*X*_3_) pulls in opposite directions—supporting yield but penalizing antioxidant activity and oxidation—so the best compromise falls where time is long enough to recover oil yet not so long that quality erodes. The overlaid desirability functions and red optima therefore converge on moderate energy with high solvent and a longer, but controlled, time window, aligning with a practical operating point around 60% energy, 20 mL/g, and 30 min. In Figure 5B, VIP scores highlight which levers matter most: *X*_3_ and *X*_2_ dominate (above the 0.8 threshold), *X*_1_’s curvature (*X*_1_^2^) and select interactions add secondary influence, and the block term shifts baselines between techniques without altering the ranking of drivers. The identical VIP scores for PEF and UBAE indicate that both innovative techniques warrant side-by-side comparison under the same PLS-optimized conditions (e.g., 60% energy, 20 mL/g, 30 min) to tease out their mechanistic differences in fat yield, antioxidant activity, and oxidation indices, as well as energy efficiency and scalability. Moreover, benchmarking these emerging methods against traditional cold-pressing and Soxhlet extraction will establish a comprehensive performance spectrum—from gentle mechanical pressing and exhaustive solvent extraction to precision-tuned PEF and cavitation-driven UBAE—enabling a data-driven choice of the most sustainable, cost-effective, and high-quality sesame seed oil process.

The predictive power of the PLS model is strongly supported by the excellent agreement between experimental observations and predicted values for both PEF and UBAE techniques, as presented in Table 5 below. A high correlation coefficient of 0.96 indicates a near-perfect linear relationship between measured and modeled outcomes across key parameters. Additionally, the coefficient of determination (R^2^) exceeding 0.93 for both methods confirms that over 93% of the variance in experimental data is accurately explained by the model, highlighting its robustness. The exceptionally low *p*-value (<0.0001) further validates the statistical significance of this alignment, demonstrating that the discrepancies between actual and predicted results are negligible and unlikely to be due to random variation. Taken together, these metrics confirm that the PLS model is a reliable and precise tool for capturing the complex interplay of extraction variables and predicting sesame seed oil quality with confidence.

### 3.6. Benchmarking Against Conventional Methods

#### 3.6.1. Fat Yield, Antioxidant Activity, and Oxidation Indices

Having established the internal optimization patterns, we benchmarked PEF and UBAE against conventional methods (cold-pressing, Soxhlet) to contextualize their performance in terms of yield, antioxidant activity, and oxidative stability. Based on the results presented in Table 5, the extraction method had a significant impact on both the yield and quality of sesame seed oil. Soxhlet extraction produced the highest fat yield (55.65%) but was associated with the weakest antioxidant activity and the poorest oxidative stability, as reflected by elevated peroxide value, TBARS, *p*-anisidine value, and Totox value. By contrast, cold-pressed and PEF-extracted oils retained significantly stronger antioxidant potential (DPPH values of approximately 19 µmol TEAC/kg oil) and exhibited lower levels of lipid peroxidation. Cold-pressing, in particular, achieved the lowest TBARS and Totox values, making it the most effective method for preserving oil integrity. UBAE displayed an intermediate profile, combining relatively high fat recovery with moderate oxidation indices. Overall, non-thermal and milder extraction methods such as cold-pressing and PEF proved superior in maintaining oil quality, whereas Soxhlet, despite its efficiency in fat yield, resulted in extensive degradation and reduced nutritional value. These findings underscore the inherent trade-off between yield and oxidative stability, emphasizing that extraction techniques must be selected not only for efficiency but also for their ability to preserve nutritional and functional quality in edible oil processing.

Soxhlet extraction significantly enhances sesame oil yield compared to cold pressing, as continuous solvent circulation and elevated temperature improve mass transfer and recover lipids remaining in the seed matrix [41,42]. However, this advantage comes at the expense of oil quality. Soxhlet oils are typically darker due to pigment co-extraction and show higher free fatty acid levels, likely from partial hydrolysis or oxidation, while heat-sensitive lignans may degrade, reducing bioactive content [41,43,44]. Hamitri-Guefri et al. [41] reported yields of ~48% for Soxhlet versus 32% for cold pressing, attributing the difference to the solvent’s ability to overcome lipid–matrix interactions. In their study, peroxide values (PV) were 8.7 and 9.6 mEq O_2_/kg oil for Soxhlet and cold-pressed oils, respectively. Similarly, Manahi et al. [45] found higher PV in solvent-extracted oil (16.95 mEq O_2_/kg) compared with unpurified cold-pressed oil (12.5 mEq O_2_/kg). These findings confirm that while Soxhlet maximizes quantitative recovery, it compromises oxidative stability and nutritional integrity.

In this study, Soxhlet extraction was conducted for 300 min, corresponding to the conventional exhaustive extraction duration recommended for oilseed matrices [46]. This extended time was intentionally selected to represent the maximum theoretical recovery of total extractable lipids, thereby serving as a benchmark for comparing the efficiency of the non-thermal PEF and UBAE processes. In contrast, the PEF and UBAE treatments were optimized for yield, quality, and minimal thermal load, resulting in substantially shorter extraction durations (10–30 min). The unequal durations therefore reflect a deliberate comparison between an exhaustive conventional reference and energy-efficient optimized non-thermal methods, rather than equivalent time frames. Previous reports indicate that Soxhlet extraction at 60–90 min achieves over 85–90% of total recoverable sesame oil [47,48,49], suggesting that while shorter durations could slightly reduce total yield, the relative ranking of methods (Soxhlet > UBAE > PEF > cold press) would remain unchanged.

When the DPPH scavenging activity was measured, it was found that the ability of both the Soxhlet extracted sesame oil and cold pressed extracted oil to scavenge the DPPH radical was 25% [41]. In another study, where different cold-pressed extracted vegetable oils where tested, different radical scavenging activities were measured. Flaxseed, grape, maize, peanut, pumpkin, rapeseed, soybean, sunflower and olive oil had 1.01, 1.43, 2.30, 0.45, 0.95, 0.80, 1.75, 1.11 and 1.29 TEAC (mmol/L) radical scavenging activity, respectively [50]. Overall, Soxhlet extraction favors high yield but may compromise color, oxidative stability, and functional constituents. Thus, extraction method depends on whether the priority is maximum recovery or preservation of nutritional and bioactive properties.

The superior preservation of radical-scavenging compounds under PEF-assisted extraction is attributed to selective electroporation of cell membranes, which enhances permeability and facilitates antioxidant release with minimal thermal stress, thereby protecting heat-sensitive bioactives [51,52]. In contrast, UBAE improves extraction yield through acoustic cavitation, where bubble collapse generates localized shear forces that disrupt cell walls and enhance solvent diffusion [35,36]. These distinct mechanisms—electroporation-driven compound release (PEF) and cavitation-induced matrix disruption (UBAE)—explain the observed differences in extraction efficiency and antioxidant retention (Figure 6).

Sarkis et al. [53] investigated the application of PEF for oil extraction from sesame seeds. The sesame seeds analyzed contained 51.58 oil and ~3.6 moisture, values typical of high-quality sesame. Considering that mechanical pressing of untreated seeds generally achieves up to 80% oil yield, all pre-treatments applied significantly improved recovery compared to the control. PEF treatment at 40 kJ kg^−1^ increased oil yield by 4.9% relative to the untreated sample and by 10.2% compared with the experimental control, although grinding remained the most effective method. It is important to mention that the samples were subjected to pretreatments like soaking and drying prior to PEF application. Earlier reports of PEF enhanced oil extraction from maize (+2.9%), olives (+7.4%), and rapeseed (+9%) [54,55]. The gain observed in sesame demonstrates that PEF is effective even at moderate energy inputs, offering a practical balance between yield improvement and energy consumption. High-performance liquid chromatography confirmed that the lignan profile (sesamin, sesamolin) and other nutritional components were not altered by PEF, supporting previous evidence of preserved oil quality and enrichment in bioactive compounds such as phytosterols and tocopherols [53,54,55].

Bahramian et al. [56] studied the effect of ultrasound process on the extraction yield and the quality of sesame oil. Ultrasound-assisted extraction demonstrated that sesame oil is resistant to oxidation during processing. Extending ultrasonic treatment to 15 min caused no significant changes in PV or CD value, while longer exposures (15–60 min) led to modest but significant increases, more pronounced for PV in *n*-hexane and CD value in petroleum ether. Similar trends—initial peroxide formation followed by decomposition of unstable hydroperoxides—have been reported for papaya seed oil [57] and rice bran oil [58]. The high oxidative stability of sesame oil, despite its ~80% unsaturated fatty acids, is attributed to sesamin, sesamolin, and abundant tocopherols [59,60]. Maintaining a frequency of 50 kHz and temperature of 25 °C likely limited cavitation intensity and prevented oxidation, consistent with observations in peanut oil where higher frequencies reduced bubble size and heat generation [61]. Fatty acid composition remained largely unchanged, with only a minor (~1%) reduction in PUFA and no significant alterations in saturated fatty acids or MUFA, consistent with previous reports for sunflower [62], rice bran [63], and tea oils [64].

Overall, these results demonstrate that both PEF and ultrasound provide gentle, energy-efficient pre-treatments that enhance extraction efficiency while preserving the chemical integrity and oxidative stability of high-value sesame oil.

#### 3.6.2. Fatty Acid Composition and Nutritional Quality Indices

According to the results in Table 6, the saturated fatty acids (SFAs) were remarkably consistent across PEF, UBAE, and cold-pressed oils (∑SFA ≈ 22.3–22.8%), with Soxhlet showing a slight increase to 23.4%. The C22:0 fraction dropped significantly in cold-pressed oil (0.19%) compared to the other methods (0.38–0.46%), suggesting that milder treatment preserves very-long-chain SFAs differently. Monounsaturated fatty acids (MUFAs), dominated by oleic acid (≈44%), and polyunsaturated fatty acids (PUFAs, ≈33%) remained virtually unchanged regardless of extraction technique. This uniformity indicates that none of the methods selectively enrich or deplete these key fatty acids.

The PUFA:SFA ratio peaked in UBAE oil (1.49), dipped in cold-pressed (1.44), and was lowest in Soxhlet (1.40), implying that ultrasound assistance may slightly improve nutritional balance. The ω-6/ω-3 ratio hovered around 30:1 for all samples, consistent with sesame’s known profile but above the ideal dietary target.

To evaluate the nutritional quality of these profiles, several indices were calculated as follows:

The Atherogenicity Index (AI) reflects the potential to promote atheroma formation [65]. Lower values indicate a more favorable lipid profile. All oils showed low AI values (~0.11), with Soxhlet slightly higher.(11)$$\mathrm{AI}=\frac{\left[C12:0+\left(4\times C14:0\right)+C16:0\right]}{\sum \mathrm{UFA}}$$

The Thrombogenic Index (TI) estimates the tendency to form clots [65]. All oils had low TI values (0.53–0.56), with Soxhlet again slightly less favorable.(12)$$\mathrm{TI}=\frac{\left(C14:0+C16:0+C18:0\right)}{\left[(0.5\times \sum \mathrm{MUFA})+(0.5\times \sum \omega \text{-}6)+(3\times \sum \omega \text{-}3)+(\omega \text{-}3/\omega \text{-}6)\right]}$$

The Hypocholesterolemic/Hypercholesterolemic ratio (H/H) estimates the balance between cholesterol-lowering and cholesterol-raising fatty acids. Higher values indicate a more favorable lipid profile [66]. In this study, H/H values ranged from ~8.7 to 8.9, with Soxhlet oil marginally lower than the others.(13)$$H/H=\frac{(C18:1+\sum \mathrm{PUFA})}{\left(C12:0+C14:0+C16:0\right)}$$

The Health-Promoting Index (HPI) complements AI, with higher values reflecting healthier profiles [67]. Sesame oils showed favorable HPI values, again slightly reduced in Soxhlet.(14)$$\mathrm{HPI}=\frac{\sum \mathrm{UFA}}{\left[C12:0+\left(4\times C14:0\right)+C16:0\right]}$$

The Calculated Oxidizability Value (COX) estimates susceptibility to oxidation [68]. All oils had similar COX values (~3.9–4.0), indicating comparable oxidative sensitivity.(15)$$\mathrm{COX}=\frac{1\left(18:1,\;\%\right)+10.3\left(18:2,\;\%\right)+21.6\left(18:3,\;\%\right)}{100}$$

Overall, the indices confirm that sesame oil, regardless of extraction method, has a favorable fatty acid profile with strong cardioprotective potential. Soxhlet extraction produced slightly less favorable values, consistent with its harsher processing conditions.

#### 3.6.3. ATR–FTIR Spectra Analysis

ATR-FTIR analysis showed that all four extraction methods—PEF, UBAE, cold-pressing, and Soxhlet—produce oils with virtually identical spectral signatures, indicating preservation of the same chemical functionalities. Figure 7 and Table 7 showed that the key stretching bands at 3005 cm^−1^ (=C–H), 2922 and 2853 cm^−1^ (CH_2_ asymmetrical and symmetrical), and 1746 cm^−1^ (ester C=O) exhibited absorbance intensities in the narrow ranges of 0.137–0.146, 0.998–1.034, 0.714–0.762, and 1.110–1.161, respectively, with no significant differences (*p* > 0.05). Likewise, fingerprint-region peaks at 988, 966, 912, and 721 cm^−1^—reflecting *cis*-C=C stretching and CH_2_ rocking—showed consistent mean values (0.18–0.195, 0.184–0.195, 0.147–0.157, and 0.553–0.600, respectively) across treatments. The absence of any statistically significant shifts or intensity changes confirms that neither PEF nor UBAE introduces structural alterations compared to conventional cold-pressing and exhaustive Soxhlet extraction. This uniformity underscores that the chemical integrity of sesame oil is maintained regardless of whether non-thermal (PEF, UBAE) or thermal-solvent (Soxhlet) techniques are employed.

Lipid oxidation products are increasingly recognized as mediators of oxidative stress, inflammation, and disease pathogenesis. Oxidation indices such as peroxide value (PV), thiobarbituric acid reactive substances (TBARS), *p*-anisidine value (*p*-AV), and Totox value provide useful indicators of lipid stability and potential health risks [69]. PV measures primary oxidation products (peroxides). Elevated PV correlates with higher dietary oxidative load and increased cardiovascular risk, largely via ROS generation and vascular damage [70,71]. Consuming diets abundant in fresh fruits, vegetables, and whole grains, while limiting processed foods, reduces these health risks through enhanced antioxidant intake [72]. TBARS are markers of PUFA oxidation and systemic oxidative stress. High TBARS levels are linked to hypertension, hyperlipidemia, and increased cardiovascular events, independent of traditional risk factors. Consumption of diets rich in unhealthy fats and deficient in antioxidants may elevate the overall dietary oxidative burden [73,74]. *p*-Anisidine value measures secondary oxidation products (aldehydes). Elevated values indicate rancidity and poorer oil quality, associated with vascular dysfunction and atherosclerosis development [75]. Repeated heating of oils exacerbates these risks. Totox, a composite index (2PV + *p*-AV), estimates overall oxidative status. High Totox values suggest significant oxidative stress potential and increased cardiovascular risk. Alarmingly, many commercial omega-3 supplements exceed recommended oxidation thresholds [76].

These findings collectively underscore the interplay between chronic oxidative stress, lipid peroxidation, and inflammation as central mechanisms in metabolic disorders, including obesity, diabetes, cardiovascular disease, and neurodegeneration [77,78]. Lipid oxidation products (LOPs), both dietary and endogenous, amplify these processes in a self-perpetuating cycle, promoting atherosclerosis and other pathologies [79,80,81]. Conversely, diets high in polyphenol-rich, minimally processed foods (e.g., fruits, vegetables, whole grains, tea, wine) offer protective antioxidant and anti-inflammatory benefits. Preventive strategies also include minimizing dietary exposure to oxidized fats by limiting processed or repeatedly heated oils, reducing storage times, and incorporating natural antioxidants in food formulations [82,83].

Recent evidence emphasizes that dietary patterns, rather than isolated nutrients, exert the strongest influence on health outcomes. For example, adherence to the Mediterranean diet has been associated with reduced oxidative stress and cardiovascular risk [73]. Consequently, contemporary approaches to cardiovascular disease management increasingly integrate nutraceutical interventions alongside pharmacological therapies. Health indices of fatty acids and oxidation markers provide useful frameworks for evaluating its nutritional value, though they should be interpreted with caution. Future research should focus on elucidating the mechanisms of sesame oil phytochemicals, validating their clinical efficacy, and exploring synergistic effects within broader dietary patterns. Expanding knowledge in this area will enhance the development of functional foods and nutraceuticals and provide new avenues for preventing and managing chronic diseases [72].

#### 3.6.4. Scalability and Energy Consumption for the Optimal PEF and UBAE Conditions

It should be noted from the outset that the apparent laboratory energy demands are inflated due to the very small sample masses processed and therefore do not reflect the true energy efficiency of these technologies. Under the experimental conditions, PEF treatment at 1 kV/cm for 10 min yielded 45.65% oil with an apparent energy demand of 1.68 × 10^4^ kJ/kg oil, while UBAE at 132 W for 30 min yielded 50.41% oil with 2.36 × 10^5^ kJ/kg oil. These values are artefacts of laboratory scale operation, where fixed power input, heat losses, and non-continuous operation greatly exaggerate the specific energy when only a few grams of sample are processed [84,85,86,87]. At pilot or industrial scale, these effects are mitigated through optimized chamber geometry, pulse modulation, and continuous flow, reducing specific energy by one to two orders of magnitude. Indeed, scaled-up PEF systems typically operate within 100–120 kJ/kg oil, and continuous ultrasound extractors within 200–300 kJ/kg oil, values that are consistent with the literature [36]. In contrast, conventional cold pressing (≈800–1500 kJ/kg) and Soxhlet extraction (>5000 kJ/kg) remain considerably more energy-intensive [88,89]. These findings demonstrate that, despite the inflated laboratory figures, both PEF and UBAE are scalable, energy-efficient, and industrially viable non-thermal strategies for sustainable sesame oil recovery.

### 3.7. Human Health Relevance and Future Perspectives

Finally, the implications of these findings were considered in relation to human health, highlighting how optimized non-thermal extraction can support functional food development. Beyond extraction efficiency, the nutritional and oxidative quality of edible oils is a critical determinant of their role in health-oriented dietary patterns. In this study, the optimized non-thermal methods (PEF and UBAE) produced oils with enhanced radical-scavenging activity, low levels of primary and secondary oxidation products, and preserved fatty acid profiles. From a human health perspective, such oils may contribute to lowering the dietary oxidative burden, supporting vascular function, and protecting polyunsaturated fatty acids from peroxidation in vivo.

As shown in Section 3.6.2, all sesame oils exhibited low AI and TI values and favorable H/H and HPI ratios, confirming their cardioprotective potential. Soxhlet oil was slightly less favorable, while PEF and cold-pressed oils maintained the most balanced profiles. These findings suggest that non-thermal methods not only preserve bioactive compounds but also sustain lipid profiles aligned with cardiovascular health.

Importantly, the chemical integrity of sesame oil was preserved irrespective of extraction technique, confirming that lignans, tocopherols, and polyunsaturated fatty acids remain intact [7,90]. This preservation underpins the oil’s potential clinical efficacy in managing oxidative stress, inflammation, and lipid-related disorders.

Collectively, these results highlight sesame oil as a promising functional food ingredient, owing to its rich content of bioactive compounds with antioxidant, cardioprotective, anti-inflammatory, and neuroprotective properties [4,91]. Non-thermal approaches such as PEF and UBAE therefore represent complementary and scalable strategies for producing high-value edible oils aligned with human health priorities.

## 4. Conclusions

This study demonstrates that extraction technique and parameter optimization decisively shape both yield and health-relevant quality attributes of sesame seed oil. Using RSM and PLS modeling, we identified complementary, non-thermal strategies: pulsed electric field (PEF) at 100% energy, 17 mL/g, 10 min maximized antiradical activity (36.0 μmol TEAC/kg oil) while maintaining low oxidative markers, and ultrasound bath-assisted extraction (UBAE) at 60% energy, 20 mL/g, 30 min delivered the highest fat recovery (59.0%) with oxidative stability suitable for storage and processing. Benchmarking against cold-pressing and Soxhlet confirmed that cold-pressing and PEF retained higher antioxidant potential (~19 μmol TEAC/kg) and markedly lower Totox values (<230) than Soxhlet (>560), underscoring the advantage of milder, green processing.

A limitation of the Soxhlet extraction applied in this study compared with the non-conventional extraction techniques used is the exhaustive character of this method. The prolonged extraction time (300 min) may overestimate its yield in relation to the optimized non-thermal PEF and UBAE processes. Sensitivity analyses and literature reports indicate that shorter Soxhlet durations (60–90 min) yield comparable extraction trends.

From a human health perspective, oils with elevated radical-scavenging capacity and low primary/secondary oxidation products may contribute to reducing dietary oxidative load, supporting vascular health, and protecting polyunsaturated fatty acids from peroxidation in vivo. These attributes are particularly relevant for functional foods aimed at cardiovascular risk reduction, oxidative stress mitigation, and healthy aging.

A limitation of the present work is that it did not directly quantify key sesame bioactives such as lignans (sesamin, sesamolin, sesamol) and tocopherols, which are strongly implicated in antioxidant capacity and associated health benefits. Future studies will target these compounds, assess their in vitro bioaccessibility, and, where feasible, examine post-ingestion biomarkers of oxidative stress and lipid metabolism in human studies. Incorporating such data will more fully substantiate the link between optimized extraction conditions, bioactive preservation, and potential health outcomes.

Practically, PEF at high-energy/short-time and UBAE at low-energy/short-time offer scalable, energy-efficient routes to high-value, bioactive-rich oils, expanding the processing toolbox for health-oriented edible oil products. The quantification of lignans and tocopherols, followed by in vitro bioaccessibility or human intervention studies, is the next essential step to link optimized extraction parameters to verified health benefits.

## Figures and Tables

**Figure 1 foods-14-03653-f001:**
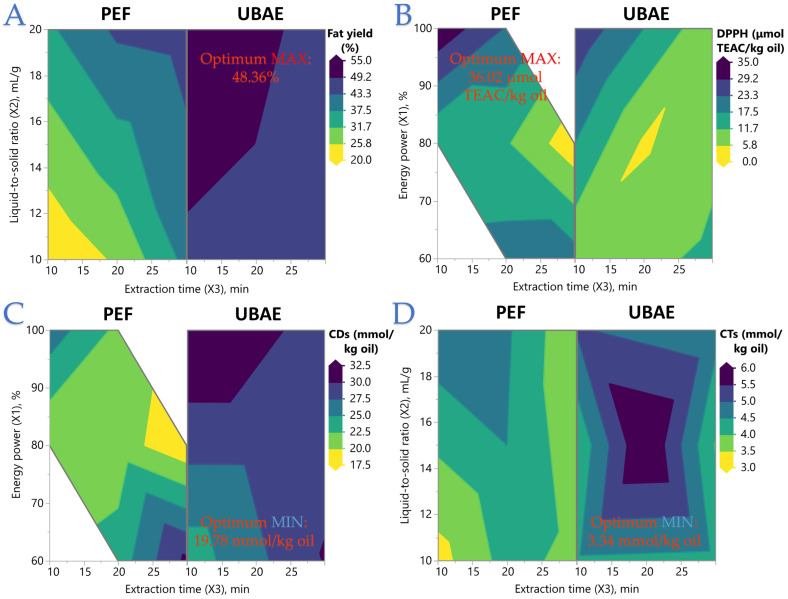
Contour plots illustrating the influence of processing parameters on key extraction responses under Pulsed Electric Field (PEF) and Ultrasound Bath-Assisted Extraction (UBAE) treatments. (**A**) Fat yield (%), (**B**) DPPH radical scavenging activity (µmol TEAC/kg oil), (**C**) conjugated dienes (CDs, mmol/kg oil), and (**D**) conjugated trienes (CTs, mmol/kg oil) as functions of extraction time (*X*_3_), liquid-to-solid ratio (*X*_2_), and energy power (*X*_1_). Optimum values derived from the response surface methodology (RSM) are directly labeled on each plot. Unified color scales, font styles, and axis formats were applied to enhance readability and facilitate comparison between extraction techniques.

**Figure 2 foods-14-03653-f002:**
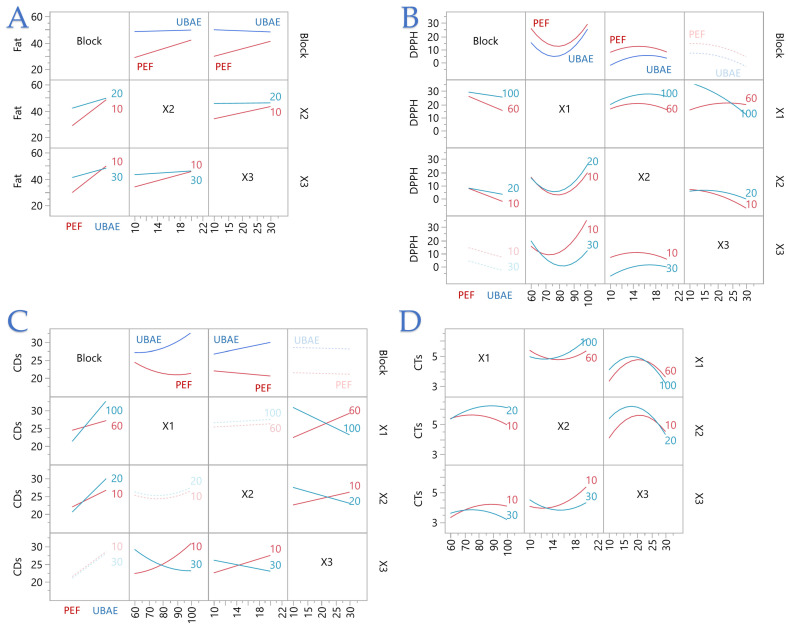
Interaction plots illustrating the effects of extraction parameters on (**A**) fat yield, (**B**) DPPH radical scavenging activity, (**C**) conjugated dienes (CDs), and (**D**) conjugated trienes (CTs) under PEF and UBAE conditions. Red and blue lines correspond to PEF and UBAE, respectively. Each plot displays the pairwise interactions between energy power (*X*_1_), liquid-to-solid ratio (*X*_2_), and extraction time (*X*_3_) generated by the RSM.

**Figure 3 foods-14-03653-f003:**
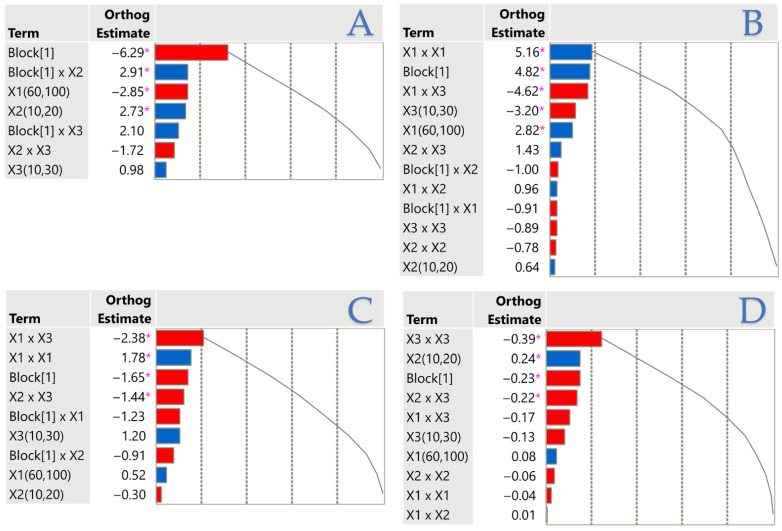
Pareto plots of standardized effect estimates for (**A**) fat content, (**B**) DPPH antiradical activity, (**C**) conjugated dienes, and (**D**) conjugated trienes. Blue bars indicate positive coefficients; red bars indicate negative coefficients. Terms significant at *p* < 0.05 are marked with an asterisk.

**Figure 4 foods-14-03653-f004:**
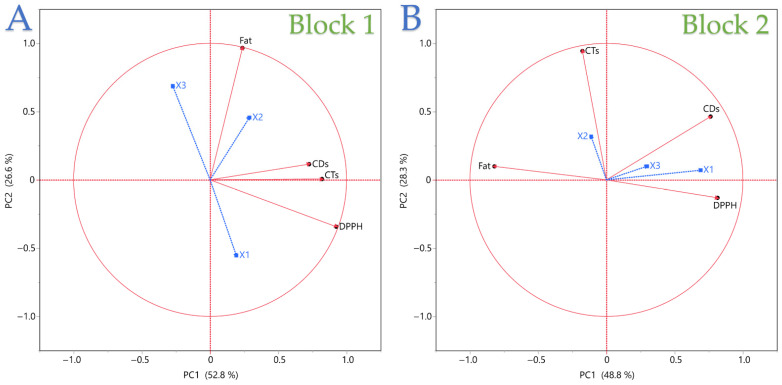
Principal component analysis biplots of extraction parameters and response variables for (**A**) PEF (Block 1) and (**B**) UBAE (Block 2). Each X variable is presented with a blue color.

**Figure 5 foods-14-03653-f005:**
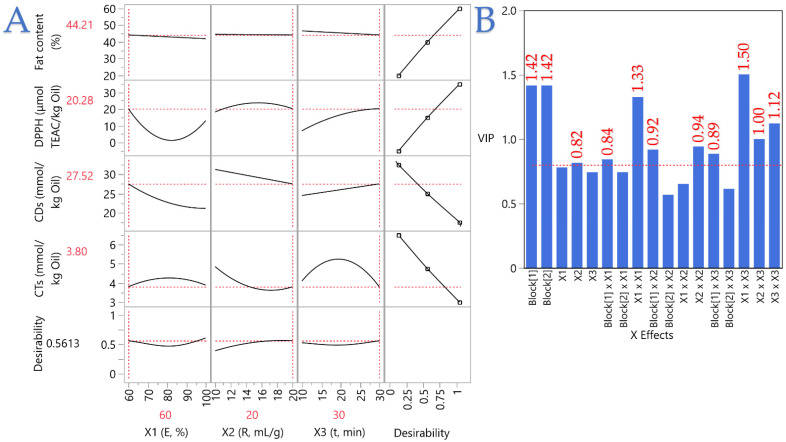
PLS prediction profiler and VIP scores for extraction optimization under PEF (Block 1) and UBAE (Block 2) conditions. (**A**) PLS prediction profiles with overlaid desirability functions, showing how fat content, DPPH activity, conjugated dienes, conjugated trienes, and overall desirability vary with energy power (*X*_1_), liquid-to-solid ratio (*X*_2_), and extraction time (*X*_3_). Red dashed lines mark the optimum response values. (**B**) Variable Importance in Projection (VIP) scores for all main, quadratic, and interaction effects; the red dashed line denotes the 0.8 threshold for significance.

**Figure 6 foods-14-03653-f006:**
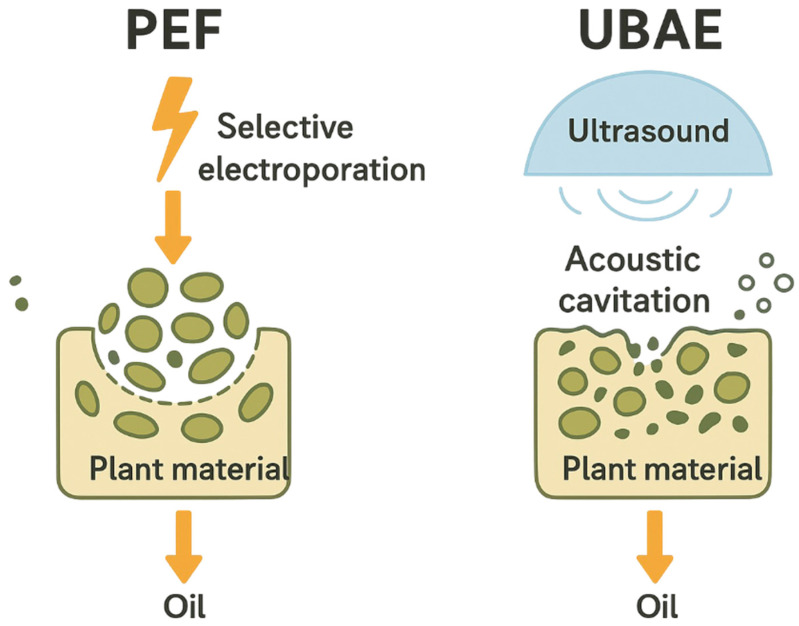
Antioxidant compounds’ release mechanisms for PEF and UBAE treatments.

**Figure 7 foods-14-03653-f007:**
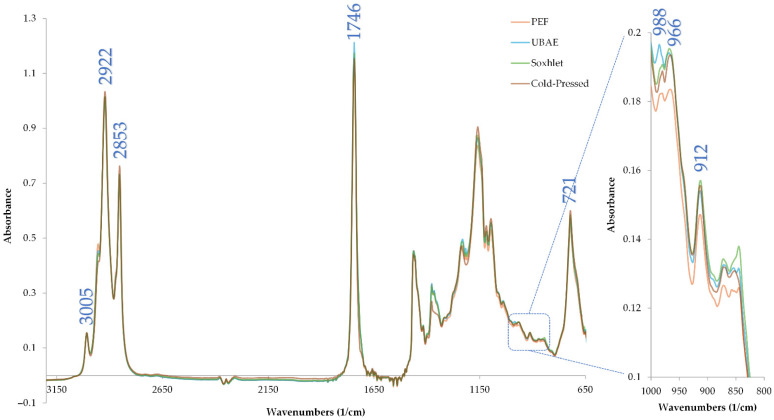
ATR-FTIR spectra of sesame seed oil samples extracted by PEF, UBAE, cold-pressing, and Soxhlet. Absorbance is plotted versus wavenumber (cm^−1^).

**Table 1 foods-14-03653-t001:** The actual and coded levels of the independent variables that were used to optimize the process.

Independent Variables	Coded Units	Coded Variable Level		
		−1	0	1
Energy power * (*E*, %)	*X* _1_	60	80	100
Liquid-to-solid ratio (*R*, mL/g)	*X* _2_	10	15	20
Extraction time (*t*, min)	*X* _3_	10	20	30

* 60%, 80% and 100% Energy power corresponds to 0.6, 0.8 and 1 kV/cm for PEF treatment and 132, 176 and 220 W for UBAE treatment, respectively.

**Table 2 foods-14-03653-t002:** Experimental results showing the effects of the three independent variables on the corresponding dependent responses.

Design Point	Independent Variables				Actual Responses *			
	Block (Technique)	*X*_1_ (*E*, %)	*X*_2_ (*R*, mL/g)	*X*_3_ (*t*, min)	Fat	DPPH	CDs	CTs
1	1 (PEF)	−1 (60)	1 (20)	0 (20)	45 ± 3.2	20.7 ± 1.2	24.3 ± 0.9	4.8 ± 0.4
2	1 (PEF)	1 (100)	1 (20)	−1 (10)	35.3 ± 1.6	33.7 ± 1.9	26.9 ± 1.6	4.8 ± 0.3
3	1 (PEF)	0 (80)	−1 (10)	−1 (10)	21.7 ± 0.8	12.7 ± 0.4	19.2 ± 0.4	3.5 ± 0.3
4	2 (UBAE)	−1 (60)	0 (15)	−1 (10)	45.6 ± 3.4	13.1 ± 0.3	27.3 ± 1.9	4.6 ± 0.2
5	2 (UBAE)	1 (100)	1 (20)	1 (30)	36.6 ± 0.7	12.9 ± 0.7	30.3 ± 1	5 ± 0.3
6	1 (PEF)	1 (100)	−1 (10)	0 (20)	30.5 ± 1.6	23.6 ± 1.5	23.1 ± 1.5	4.1 ± 0.1
7	1 (PEF)	−1 (60)	−1 (10)	1 (30)	37.9 ± 2	20 ± 1.2	31.9 ± 1	3.9 ± 0.2
8	2 (UBAE)	−1 (60)	−1 (10)	−1 (10)	50.5 ± 3.8	5.2 ± 0.3	19 ± 0.6	4 ± 0.2
9	2 (UBAE)	0 (80)	0 (15)	0 (20)	53.4 ± 1.5	6.9 ± 0.4	31.5 ± 1.4	6.2 ± 0.4
10	1 (PEF)	0 (80)	0 (15)	0 (20)	32.9 ± 2.1	9.2 ± 0.6	18.7 ± 0.5	4.2 ± 0.2
11	2 (UBAE)	1 (100)	−1 (10)	1 (30)	45.1 ± 2.9	0.1 ± 0	28.5 ± 1	4.6 ± 0.2
12	1 (PEF)	1 (100)	1 (20)	−1 (10)	35.9 ± 0.8	33.5 ± 2	26.3 ± 0.7	4.9 ± 0.4
13	2 (UBAE)	1 (100)	0 (15)	1 (30)	52.7 ± 2.1	8.9 ± 0.5	26.7 ± 2	3.6 ± 0.1
14	1 (PEF)	0 (80)	1 (20)	1 (30)	47.8 ± 2.6	2.8 ± 0.1	18.3 ± 0.8	3.5 ± 0.1
15	2 (UBAE)	−1 (60)	1 (20)	1 (30)	50.4 ± 2.4	12.5 ± 0.7	28.6 ± 1.4	4.5 ± 0.3
16	2 (UBAE)	−1 (60)	1 (20)	−1 (10)	59 ± 1.8	3.1 ± 0.1	24.5 ± 0.6	4.7 ± 0.2
17	1 (PEF)	−1 (60)	−1 (10)	1 (30)	37 ± 2.7	20.4 ± 1.4	30.1 ± 1.3	3.8 ± 0.2
18	2 (UBAE)	1 (100)	−1 (10)	−1 (10)	44 ± 1	27.1 ± 0.7	32.1 ± 2.1	4.5 ± 0.2

* Values represent the mean ± standard deviation of triplicate determinations (*n* = 3); Fat content in %; DPPH antiradical activity in μmol TEAC/kg oil; Conjugated dienes (CDs) and conjugated trienes (CTs) in mmol/kg oil.

**Table 3 foods-14-03653-t003:** Maximum predicted responses and optimum extraction conditions for the dependent variables.

Responses	Independent Variables				Desirability	Stepwise Regression
	Block (Technique)	*X*_1_ (*E*, %)	*X*_2_ (*R*, mL/g)	*X*_3_ (*t*, min)		
Fat content (%)	UBAE	60	20	30	0.6978	48.36 ± 6.59
DPPH (μmol TEAC/kg oil)	PEF	100	17	10	0.9923	36.02 ± 5.92
CDs (mmol/kg oil)	UBAE	60	10	10	0.8341	19.78 ± 4.78
CTs (mmol/kg oil)	UBAE	60	12	10	0.8885	3.34 ± 0.75

**Table 4 foods-14-03653-t004:** Pearson correlation coefficients among Fat, DPPH, CDs, and CTs under PEF (Block 1) and UBAE (Block 2) conditions.

Responses	Fat	DPPH	CDs	CTs
	Block 1			
Fat	-	−0.100	0.223	0.237
DPPH		-	0.609	0.742
CDs			-	0.233
CTs				-
	Block 2			
Fat	-	−0.478	−0.470	0.179
DPPH		-	0.456	−0.192
CDs			-	0.178
CTs				-

**Table 5 foods-14-03653-t005:** Fat yield, antioxidant capacity, and oxidative stability indices of sesame seed oils extracted by PEF, UBAE, Cold-Pressing, and Soxhlet methods.

Responses	PLS-Predicted	PEF	UBAE	Cold-Pressed	Soxhlet	Formula (Section)
Fat content (%)	44.21	45.65 ± 2.65 ^b^	50.41 ± 1.61 ^a,b^	51.78 ± 2.12 ^a^	55.65 ± 1.5 ^a^	2.4.1
DPPH (μmol TEAC/kg oil)	20.28	19.49 ± 1.35 ^a^	12.67 ± 0.84 ^b^	18.66 ± 1.16 ^a^	6.03 ± 0.18 ^c^	2.4.2
CDs (mmol/kg oil)	27.52	19.13 ± 1.24 ^c^	22.41 ± 0.94 ^b^	14.25 ± 0.68 ^d^	26.04 ± 1.41 ^a^	2.4.3
CTs (mmol/kg oil)	3.80	2.95 ± 0.18 ^b^	3.21 ± 0.19 ^b^	2.16 ± 0.11 ^c^	3.9 ± 0.25 ^a^	2.4.3
PV (mmol H_2_O_2_/kg oil)		114.07 ± 6.84 ^c^	160.85 ± 11.9 ^b^	92.08 ± 4.24 ^c^	281.69 ± 9.01 ^a^	2.5.1
TBARS (mmol MDAE/kg oil)		0.3 ± 0.01 ^c^	0.69 ± 0.04 ^b^	0.08 ± 0 ^d^	2.01 ± 0.13 ^a^	2.5.2
*p*-AV		0.5 ± 0.03 ^c^	0.66 ± 0.04 ^b^	0.41 ± 0.02 ^c^	2.3 ± 0.09 ^a^	2.5.3
Totox value		228.65 ± 13.72 ^c^	322.35 ± 23.84 ^b^	184.58 ± 8.49 ^c^	565.67 ± 18.11 ^a^	2.5.4

Data are reported as the mean of three independent replicates ± standard deviation. In each row, values carrying different superscript letters (e.g., a–d) are significantly different at *p* < 0.05.

**Table 6 foods-14-03653-t006:** Fatty acid composition and nutritional quality indices of sesame seed oils extracted by PEF, UBAE, Cold-Pressing, and Soxhlet methods.

Samples	PEF (%)	UBAE (%)	Cold-Pressed (%)	Soxhlet (%)
C16:0	8.68 ± 0.49 ^a^	8.7 ± 0.45 ^a^	8.65 ± 0.38 ^a^	8.77 ± 0.5 ^a^
C18:0	13.6 ± 0.31 ^a^	13.24 ± 0.53 ^a^	13.98 ± 0.95 ^a^	14.19 ± 1.05 ^a^
C22:0	0.38 ± 0.01 ^b^	0.38 ± 0.03 ^b^	0.19 ± 0 ^c^	0.46 ± 0.02 ^a^
∑ SFA	22.66 ± 0.81 ^a^	22.32 ± 1.01 ^a^	22.82 ± 1.34 ^a^	23.42 ± 1.57 ^a^
C18:1	44.29 ± 2.26 ^a^	44.43 ± 2.18 ^a^	44.24 ± 1.28 ^a^	43.83 ± 2.37 ^a^
∑ MUFA	44.29 ± 2.26 ^a^	44.43 ± 2.18 ^a^	44.24 ± 1.28 ^a^	43.83 ± 2.37 ^a^
C18:2 (ω-6)	31.99 ± 1.41 ^a^	32.18 ± 1.54 ^a^	31.92 ± 1.69 ^a^	31.72 ± 1.43 ^a^
C18:3 (ω-3)	1.06 ± 0.06 ^a^	1.07 ± 0.07 ^a^	1.02 ± 0.03 ^a^	1.03 ± 0.02 ^a^
∑ PUFA	33.05 ± 1.47 ^a^	33.25 ± 1.61 ^a^	32.94 ± 1.73 ^a^	32.75 ± 1.45 ^a^
∑ UFA	77.34 ± 3.72 ^a^	77.68 ± 3.79 ^a^	77.18 ± 3.01 ^a^	76.58 ± 3.82 ^a^
PUFA:SFA ratio	1.46 ± 0.01 ^a^	1.49 ± 0 ^a^	1.44 ± 0.01 ^a,b^	1.4 ± 0.03 ^b^
MUFA:PUFA ratio	1.34 ± 0.01 ^a^	1.34 ± 0 ^a^	1.34 ± 0.03 ^a^	1.34 ± 0.01 ^a^
ω-6/ω-3 ratio	30.19 ± 0.33 ^a^	30.09 ± 0.42 ^a^	31.28 ± 0.63 ^a^	30.79 ± 0.65 ^a^
AI	0.11 ± 0 ^b^	0.11 ± 0 ^b^	0.11 ± 0 ^b^	0.11 ± 0 ^a^
TI	0.54 ± 0.01 ^a,b^	0.53 ± 0 ^b^	0.55 ± 0.01 ^a,b^	0.56 ± 0.01 ^a^
H/H	8.91 ± 0.07 ^a^	8.93 ± 0.03 ^a^	8.92 ± 0.04 ^a^	8.73 ± 0.06 ^b^
HPI	8.91 ± 0.07 ^a^	8.93 ± 0.03 ^a^	8.92 ± 0.04 ^a^	8.73 ± 0.06 ^b^
COX	3.97 ± 0.18 ^a^	3.99 ± 0.2 ^a^	3.95 ± 0.19 ^a^	3.93 ± 0.18 ^a^

Data are reported as the mean of three independent replicates ± standard deviation. In each row, values carrying different superscript letters (e.g., a–c) are significantly different at *p* < 0.05.

**Table 7 foods-14-03653-t007:** ATR-FTIR absorbance intensities at characteristic wavenumbers (cm^−1^) for sesame seed oils obtained by PEF, UBAE, Cold-Pressing, and Soxhlet extraction.

Wavenumbers (cm^−1^)	PEF	UBAE	Cold-Pressed	Soxhlet
3005	0.137 ± 0.009	0.143 ± 0.007	0.146 ± 0.004	0.144 ± 0.009
2922	0.998 ± 0.052	1.014 ± 0.068	1.034 ± 0.022	1.015 ± 0.048
2853	0.714 ± 0.045	0.732 ± 0.045	0.762 ± 0.018	0.732 ± 0.032
1746	1.117 ± 0.067	1.161 ± 0.086	1.11 ± 0.064	1.129 ± 0.078
988	0.18 ± 0.007	0.195 ± 0.012	0.184 ± 0.013	0.187 ± 0.006
966	0.184 ± 0.009	0.194 ± 0.008	0.194 ± 0.013	0.195 ± 0.014
912	0.147 ± 0.004	0.154 ± 0.004	0.156 ± 0.004	0.157 ± 0.007
721	0.553 ± 0.032	0.572 ± 0.017	0.6 ± 0.016	0.585 ± 0.036

Data are mean ± SD of three independent replicates (*n* = 3). Within each row, values do not differ significantly (*p* < 0.05).

## Data Availability

The original contributions presented in this study are included in the article/Appendix A. Further inquiries can be directed to the corresponding author.

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
