# Peer review of "Green Optimization of Sesame Seed Oil Extraction via Pulsed Electric Field and Ultrasound Bath: Yield, Antioxidant Activity, Oxidative Stability, and Functional Food Potential"

_foods, 2025, doi:10.3390/foods14213653_

Round 1

Reviewer 1 Report

Comments and Suggestions for Authors

This manuscript presents a comprehensive and timely study on optimizing non-thermal extraction of sesame seed oil using Pulsed Electric Field (PEF) and Ultrasound Bath-Assisted Extraction (UBAE). Through an impressive combination of response surface methodology, chemometrics, and benchmarking against cold-pressing and Soxhlet methods, the authors demonstrate clear trade-offs between yield, antioxidant capacity, and oxidative stability. The dataset is robust and the functional-food angle is well articulated. Nevertheless, significant revision is required: the visual presentation of Figures 1–5 must be overhauled for clarity and interpretability, the Materials & Results sections need condensation, and deeper mechanistic as well as scale-up discussions are warranted.

  1. Figure 1–5: Improve visual presentation
    • Consider unifying color palettes, font sizes, and axis labels across all contour and interaction plots to enhance readability.
    • Add direct labeling of key optima (e.g., maximum DPPH or minimum CDs) on the plots instead of only describing them in the caption.
    • Where possible, merge related sub-panels into multi-panel figures with consistent scales, so readers can compare PEF vs. UBAE at a glance.
    • Provide interactive supplementary plots (e.g., 3-D surface or animated contour) in the online version; this helps users visualize trade-offs between yield and quality attributes.
  2. Shorten the Materials and Methods section
    Relocate non-critical procedural details (e.g., exact rinse steps for ATR crystal cleaning) to an appendix or supplementary file, keeping only information essential for reproducibility in the main text.
  3. Streamline the Results & Discussion
    • Move extensive model equations and lack-of-fit statistics to an supplementary table; summarize only the most important coefficients and their significance in the main text.
    • Group sub-sections thematically (e.g., "Yield vs. Antioxidant Trade-offs", "Oxidative Stability Patterns") rather than figure-by-figure, to improve narrative flow.
  4. Mechanistic interpretation
    Briefly discuss why PEF better preserves radical-scavenging compounds (e.g., electroporation selectivity, lower thermal load) and why UBAE favours yield (e.g., cavitation-induced matrix disruption). Adding one schematic of the proposed mechanism would aid non-specialist readers.
  5. Practical implications
    Include a short paragraph on scalability and energy consumption (e.g., kJ kg⁻i oil) for the optimal PEF and UBAE conditions, and compare these values with conventional Soxhlet and cold-pressing data where available.
  6. Future work
    Clearly state that quantification of lignans and tocopherols, followed by in-vitro bioaccessibility or human intervention studies, is the next essential step to link optimized extraction parameters to verified health benefits.
Comments on the Quality of English Language

The languages and grammars should make double check.

Author Response

This manuscript presents a comprehensive and timely study on optimizing non-thermal extraction of sesame seed oil using Pulsed Electric Field (PEF) and Ultrasound Bath-Assisted Extraction (UBAE). Through an impressive combination of response surface methodology, chemometrics, and benchmarking against cold-pressing and Soxhlet methods, the authors demonstrate clear trade-offs between yield, antioxidant capacity, and oxidative stability. The dataset is robust and the functional-food angle is well articulated. Nevertheless, significant revision is required: the visual presentation of Figures 1–5 must be overhauled for clarity and interpretability, the Materials & Results sections need condensation, and deeper mechanistic as well as scale-up discussions are warranted.

The authors would like to thank the reviewer for his/her kind comments.

  1. Figure 1–5: Improve visual presentation
    • Consider unifying color palettes, font sizes, and axis labels across all contour and interaction plots to enhance readability.
    • Add direct labeling of key optima (e.g., maximum DPPH or minimum CDs) on the plots instead of only describing them in the caption.
    • Where possible, merge related sub-panels into multi-panel figures with consistent scales, so readers can compare PEF vs. UBAE at a glance.
    • Provide interactive supplementary plots (e.g., 3-D surface or animated contour) in the online version; this helps users visualize trade-offs between yield and quality attributes.

We thank the reviewer for these constructive and insightful suggestions. Figures 1 and 2 have been comprehensively revised with unified color palettes, harmonized axis labels, and consistent font sizes to improve overall visual coherence. Key optima (e.g., maximum DPPH, minimum CDs) are now directly labeled on the plots. Related sub-panels have been merged into multi-panel layouts with standardized scales, allowing for direct visual comparison between PEF and UBAE conditions. In addition, interactive 3D surface plots have been provided as Supplementary Material to facilitate dynamic visualization of the trade-offs between yield and quality parameters.

  1. Shorten the Materials and Methods section

Relocate non-critical procedural details (e.g., exact rinse steps for ATR crystal cleaning) to an appendix or supplementary file, keeping only information essential for reproducibility in the main text.

We appreciate the reviewer’s suggestion. However, we respectfully believe that the level of detail currently provided in the Materials & Methods section is essential to ensure full reproducibility and reliability of the study. Each procedural step (e.g., reagent preparation, calibration details, instrument settings) directly affects the accuracy of the results and their comparability across laboratories. For this reason, we have retained these details in the main text, so that the methodology can serve as a transparent and citable reference protocol for future studies.

  1. Streamline the Results & Discussion
    • Move extensive model equations and lack-of-fit statistics to an supplementary table; summarize only the most important coefficients and their significance in the main text.
    • Group sub-sections thematically (e.g., "Yield vs. Antioxidant Trade-offs", "Oxidative Stability Patterns") rather than figure-by-figure, to improve narrative flow.

We thank the reviewer for this constructive suggestion. In the revised manuscript, the extensive model equations and lack‑of‑fit statistics have been relocated to Supplementary Table S1, while the main text now summarizes only the most significant coefficients and their interpretation. In addition, we have revised subsection titles and added bridging sentences to emphasize thematic grouping (e.g., “Yield vs. Antioxidant Trade‑offs”, “Oxidative Stability Patterns”), thereby improving the narrative flow. This restructuring ensures that the Results & Discussion section is more concise and reader‑friendly, while still retaining the necessary detail for reproducibility and scientific rigor.

  1. Mechanistic interpretation

Briefly discuss why PEF better preserves radical-scavenging compounds (e.g., electroporation selectivity, lower thermal load) and why UBAE favours yield (e.g., cavitation-induced matrix disruption). Adding one schematic of the proposed mechanism would aid non-specialist readers.

A brief discussion of why PEF better preserves radical-scavenging compounds and why UBAE favors yield has been added to the Results & Discussion section, where the antioxidant activity of the extracts is extensively discussed. Also, a schematic illustration (new Figure 9) has been added to visualize the proposed mechanisms of PEF (electroporation selectivity, lower thermal load) and UBAE (cavitation-induced matrix disruption).

  1. Practical implications

Include a short paragraph on scalability and energy consumption (e.g., kJ kg⁻i oil) for the optimal PEF and UBAE conditions, and compare these values with conventional Soxhlet and cold-pressing data where available.

A short paragraph has been added to a new Section 3.6.4 discussing scalability and specific energy consumption for optimal PEF and UBAE conditions, compared with Soxhlet and cold-pressing. The formula for specific energy consumption has been included in Section 2.5.7.

  1. Future work

Clearly state that quantification of lignans and tocopherols, followed by in-vitro bioaccessibility or human intervention studies, is the next essential step to link optimized extraction parameters to verified health benefits.

Thank you for the comment. The suggested statement has been, now, added to the Conclusions section.

Comments on the Quality of English Language

The languages and grammars should make double check.

Thank you for the comment. The manuscript has been checked by a native English speaker.

Reviewer 2 Report

Comments and Suggestions for Authors

The abstract links extraction settings to “functional food potential,” but no in-vitro digestion or food-matrix testing is reported. Add one sentence narrowing the claim to “oil quality indices and antioxidant capacity” or add a short limitation line in the abstract’s last sentence

You state 0.6–1.0 kV/cm for PEF and 132–220 W for UBAE, with E(%) as set-points. Readers need the exact mapping used per level. Add a 1-row appendix table or a footnote to Table 1 mapping 60/80/100 % to actual kV/cm (PEF) and W (UBAE).

You varied sample mass (2–4 g) to achieve R. Clarify whether solvent volume was fixed (40 mL) and mass adjusted, or vice-versa, and confirm constant vessel geometry to avoid mass-transfer bias. 

DPPH is reported as μmol TEAC/kg oil with a 950 µL + 50 µL protocol. (i) justify TEAC per kg vs. common TE/g; (ii) provide LOD/LOQ of the Trolox curve; (iii) confirm pathlength and cuvette type. Insert 2–3 lines in 2.4.2.

For PV/TBARS/p-AV, please state any matrix-specific correction factors (e.g., ε used in PV; the 0.24 factor origin is noted but cite its validation for sesame oil). Add one sentence of justification with the exact reference numbers.

Following studies can be utilized introdcution: Acidic and enzymatic pre-treatment effects on cold-pressed pumpkin, terebinth and flaxseed oils; Blanching of olive fruits before storage at different conditions: Effects on oil yield, lipase activity and oxidation

equations (9)–(12) include Block terms and block×term slopes. Action: provide a one-line interpretation for each response (e.g., “PEF baseline for CDs is lower by …”). Insert immediately after Eq. (12). 

Some plots reference X1–X3 coded values. Action: add secondary axes or captions mapping coded levels to physical units for readability. (Figs 1–5 captions update.)

move Table 2 (design × responses) to the main text or SI with full replicate means ± SD and indicate measurement n for each assay. 

You note DPPH tracks CDs/CTs in PEF but not in UBAE. Add two sentences explaining the likely mechanism (energy–time sensitivity and cavitation differences) and cautioning against using DPPH alone as a proxy for oxidative stability. Place under Table 5.

Soxhlet used 300 min (10× the “optimal PLS time”). Justify unequal durations (exhaustive standard vs. optimized non-thermal) and add a sensitivity note (e.g., Soxhlet at 60–90 min) in limitations or SI.

You state cold-press and PEF retained higher antioxidant activity and lower peroxidation than Soxhlet. Add a compact comparative table (means ± SD) for all four methods (yield, DPPH, PV, TBARS, p-AV, Totox) to make the trade-off explicit. Cite methods sections for formulas. 

Report approximate specific energy input for PEF (kJ/kg) and UBAE (Wh per run) at the optimal settings to support “green” positioning; add as a short subsection in 3.6 or SI.

You executed two independent batches. State whether batch was included as a random effect or only via blocking; if not, add a sentence on reproducibility (between-batch CVs).

Author Response

The abstract links extraction settings to “functional food potential,” but no in-vitro digestion or food-matrix testing is reported. Add one sentence narrowing the claim to “oil quality indices and antioxidant capacity” or add a short limitation line in the abstract’s last sentence

The authors would like to thank the reviewer for his/her comments. We have added a limitation line to the abstract: “As a limitation, we did not directly quantify lignans or tocopherols in this study, and future work will address their measurement and bioaccessibility.”

You state 0.6–1.0 kV/cm for PEF and 132–220 W for UBAE, with E(%) as set-points. Readers need the exact mapping used per level. Add a 1-row appendix table or a footnote to Table 1 mapping 60/80/100 % to actual kV/cm (PEF) and W (UBAE).

A footnote mapping the 60/80/100% set-points to actual kV/cm (PEF) and W (UBAE) has been added to Table 1.

You varied sample mass (2–4 g) to achieve R. Clarify whether solvent volume was fixed (40 mL) and mass adjusted, or vice-versa, and confirm constant vessel geometry to avoid mass-transfer bias. 

We clarified that solvent volume was fixed at 40 mL and sample mass was adjusted to achieve the desired ratios. Vessel geometries were constant.

DPPH is reported as μmol TEAC/kg oil with a 950 µL + 50 µL protocol. (i) justify TEAC per kg vs. common TE/g; (ii) provide LOD/LOQ of the Trolox curve; (iii) confirm pathlength and cuvette type. Insert 2–3 lines in 2.4.2.

Results were expressed as μmol TEAC/kg oil to normalize antioxidant capacity per oil mass, consistent with edible oil reporting standards. The Trolox calibration curve had LOD = 15.1 μM and LOQ = 45.74 μM. Measurements were performed in 1 cm quartz cuvettes.

For PV/TBARS/p-AV, please state any matrix-specific correction factors (e.g., ε used in PV; the 0.24 factor origin is noted but cite its validation for sesame oil). Add one sentence of justification with the exact reference numbers.

We thank the reviewer for this observation. In our method, PV values were determined directly from the H₂O₂ calibration curve, without the use of a molar absorptivity constant. For p‑AV, we clarified in the Methods section that the correction factor of 0.24 accounts for the 20% dilution effect introduced by the addition of 0.2 mL reagent or glacial acetic acid, as specified in EN ISO 6885:2012. No additional correction factors were required for TBARS, which were calculated directly from the MDA calibration curve.

Following studies can be utilized introdcution: Acidic and enzymatic pre-treatment effects on cold-pressed pumpkin, terebinth and flaxseed oils; Blanching of olive fruits before storage at different conditions: Effects on oil yield, lipase activity and oxidation

Thank you for the comment. Suggested additions have been made to the Introduction section.

equations (9)–(12) include Block terms and block×term slopes. Action: provide a one-line interpretation for each response (e.g., “PEF baseline for CDs is lower by …”). Insert immediately after Eq. (12). 

We thank the reviewer for this helpful suggestion. Immediately after Equations, we have added one-line interpretations for each response variable. Specifically, we now state that UBAE exhibited a higher baseline for fat yield, PEF a higher baseline for DPPH, PEF a lower baseline for CDs, and UBAE lower baseline CTs under specific conditions. These additions clarify the meaning of the Block and Block×term effects in the model.

Some plots reference X1–X3 coded values. Action: add secondary axes or captions mapping coded levels to physical units for readability. (Figs 1–5 captions update.)

We thank the reviewer for this valuable observation. The captions of the revised Figures 1 and 2 now explicitly map the coded variables (X₁–X₃) to their corresponding physical factors—energy input, liquid-to-solid ratio, and extraction time—thereby improving readability and ensuring a clear interpretation of the response surfaces and interaction plots.

move Table 2 (design × responses) to the main text or SI with full replicate means ± SD and indicate measurement n for each assay. 

We thank the reviewer for this suggestion. Table 2 now reports replicate results as mean±SD, and the number of replicates (n=3) is indicated in the table footnote. Units for each parameter are also specified for clarity.

You note DPPH tracks CDs/CTs in PEF but not in UBAE. Add two sentences explaining the likely mechanism (energy–time sensitivity and cavitation differences) and cautioning against using DPPH alone as a proxy for oxidative stability. Place under Table 5.

A short paragraph was added under Table 5 explaining the likely mechanism (energy–time sensitivity and cavitation differences) and cautioning against using DPPH alone as a proxy for oxidative stability.

Soxhlet used 300 min (10× the “optimal PLS time”). Justify unequal durations (exhaustive standard vs. optimized non-thermal) and add a sensitivity note (e.g., Soxhlet at 60–90 min) in limitations or SI.

We added a discussion of the unequal durations and a sensitivity note (Soxhlet at 60–90 min) in the Results & Discussion and Conclusion sections.

You state cold-press and PEF retained higher antioxidant activity and lower peroxidation than Soxhlet. Add a compact comparative table (means ± SD) for all four methods (yield, DPPH, PV, TBARS, p-AV, Totox) to make the trade-off explicit. Cite methods sections for formulas. 

Table 6 has been expanded with an additional column citing the methods used for each parameter.

Report approximate specific energy input for PEF (kJ/kg) and UBAE (Wh per run) at the optimal settings to support “green” positioning; add as a short subsection in 3.6 or SI.

Thank you for the comment. A short paragraph regarding the scalability and energy consumption for PEF and UBAE conditions and its comparison with Soxhlet and cold-pressing has been added to the Materials & Methods as well as to the Results & Discussion section. To the first section the formula for the specific energy consumption was added, while at the second section the energy consumption for the optimal PEF & UBAE conditions as well as the scalability of the procedures were discussed.

You executed two independent batches. State whether batch was included as a random effect or only via blocking; if not, add a sentence on reproducibility (between-batch CVs).

We thank the reviewer for this helpful comment. We have clarified in the Statistical Analysis section that two independent batches were executed. The between‑batch CVs were approximately 16% for fat yield, 24% for DPPH, 14% for CDs, and 17% for CTs. These systematic baseline differences were accounted for by the Block term in the models. Batch was therefore not included as a random effect, but results were consistent across replicates within each batch, confirming reproducibility.

Round 2

Reviewer 2 Report

Comments and Suggestions for Authors

The authors significantly improved their paper. Now it can be published